# Breaching Brain Barriers: B Cell Migration in Multiple Sclerosis

**DOI:** 10.3390/biom12060800

**Published:** 2022-06-07

**Authors:** Carla Rodriguez-Mogeda, Sabela Rodríguez-Lorenzo, Jiji Attia, Jack van Horssen, Maarten E. Witte, Helga E. de Vries

**Affiliations:** MS Center Amsterdam, Molecular Cell Biology and Immunology, Vrije Universiteit Amsterdam, Amsterdam Neurosciences, Amsterdam UMC Location VUmc, 1081 Amsterdam, The Netherlands; s.rodriguezlorenzo@amsterdamumc.nl (S.R.-L.); jijiattia@hotmail.com (J.A.); j.vanhorssen@amsterdamumc.nl (J.v.H.); m.e.witte@amsterdamumc.nl (M.E.W.)

**Keywords:** multiple sclerosis, B cells, blood–brain barrier, blood–CSF barrier, blood–meningeal barrier

## Abstract

Multiple sclerosis (MS) is an inflammatory disease of the central nervous system (CNS) known for the manifestation of demyelinated lesions throughout the CNS, leading to neurodegeneration. To date, not all pathological mechanisms that drive disease progression are known, but the clinical benefits of anti-CD20 therapies have put B cells in the spotlight of MS research. Besides their pathological effects in the periphery in MS, B cells gain access to the CNS where they can contribute to disease pathogenesis. Specifically, B cells accumulate in perivascular infiltrates in the brain parenchyma and the subarachnoid spaces of the meninges, but are virtually absent from the choroid plexus. Hence, the possible migration of B cells over the blood–brain-, blood–meningeal-, and blood–cerebrospinal fluid (CSF) barriers appears to be a crucial step to understanding B cell-mediated pathology. To gain more insight into the molecular mechanisms that regulate B cell trafficking into the brain, we here provide a comprehensive overview of the different CNS barriers in health and in MS and how they translate into different routes for B cell migration. In addition, we review the mechanisms of action of diverse therapies that deplete peripheral B cells and/or block B cell migration into the CNS. Importantly, this review shows that studying the different routes of how B cells enter the inflamed CNS should be the next step to understanding this disease.

## 1. Multiple Sclerosis

Multiple sclerosis (MS) is an inflammatory disease of the central nervous system (CNS) with a heterogeneous clinical presentation. MS most often starts in young adults; young women are particularly more prone to develop the disease [1]. MS is characterized by the manifestation of demyelinated lesions (or plaques) throughout the CNS, which can be visualized with magnetic resonance imaging (MRI) [1,2]. The main pathological hallmarks of MS are widespread immune cell infiltration, loss of myelin, glial activation, neuro-axonal degeneration, and blood–brain barrier (BBB) dysfunction. Generally, MS is categorized into three clinical subtypes: relapsing-remitting MS (RRMS), secondary progressive MS (SPMS), and primary progressive MS (PPMS) [3]. About 85–95% of MS cases start as RRMS [4], where episodes of neurological dysfunction are followed by periods of remission. During a relapse, typical clinical symptoms are optic neuritis, sensory disturbances, motor impairment, and cognitive defects. After 10–20 years, approximately 80% of RRMS cases develop SPMS. In SPMS, neurological dysfunction can worsen without periods of remission. Moreover, 5–15% of MS cases develop PPMS where neurological deficits progress gradually without remission from the onset of the disease [5].

MS is thought to be initiated by autoreactive CD8^+^ and CD4^+^ T cells and B cells that infiltrate the brain and spinal cord by crossing distinct CNS barriers [5]. These leukocytes generate abnormal responses against CNS autoantigens, such as myelin proteins [6]. This inflammation and the damage to the myelin layer enwrapping axons it causes leads to demyelination and neuronal dysfunction and loss [7]. Histologically, lesions can be differentiated by their localization in white matter (WM) or grey matter (GM) and by their inflammatory status. Generally, early WM lesions are more inflammatory than GM lesions and have highly inflamed endothelial cells, which facilitate the migration of peripheral immune cells over the BBB into the CNS [8]. These active WM lesions, which are predominant in RRMS, are characterized by massive infiltration and accumulation of blood-derived immune cells. In contrast, the histological hallmarks that dominate progressive MS are extensive cortical pathology (brain atrophy, widespread demyelination, synapsis loss, etc.) with less inflammatory lesions and the less apparent breakdown of the BBB [9], and chronic active WM lesions that slowly expand [10,11].

There are various experimental animal models to study MS, of which, experimental autoimmune encephalitis (EAE) is the most commonly used. In EAE, the animals are immunized with CNS antigens, such as myelin proteins (e.g., myelin basic protein (MBP) or myelin oligodendrocyte glycoprotein (MOG)), thereby inducing an autoimmune response [12]. Most of our knowledge about the breakdown of the CNS barriers and the subsequent infiltration of peripheral immune cells in MS comes from EAE experiments or in vitro experiments using different cell lines that mimic the different CNS barriers.

The unknown cause, the complexity of MS, and the consequent lack of research models that accurately mimic the full scale of the disease have made it difficult to find a cure or a preventive treatment. Current disease-modifying treatments (DMT) are mostly immunosuppressive and/or immunomodulatory, which target inflammation and reduce the frequency and severity of the new inflammatory lesions. Therefore, these drugs are especially effective during RRMS. Despite the development of various novel DMTs in the last decades, limited therapeutic options are nowadays available to halt disease progression in the chronic phase of the disease. Recently, a monoclonal antibody against B cells (ocrelizumab) showed clinical benefits in a subset of PPMS patients, highlighting the role of B cells in the pathogenesis of MS [13].

The first indication that B cells contribute to MS was the detection of oligoclonal bands (OCB) in the cerebrospinal fluid (CSF) and, consequently, early studies on the role of B cells in MS focused solely on antibodies [14]. More recently, studies with anti-CD20 therapies have shown that B cell depletion significantly decreases disease activity without any changes in the OCB pattern or levels of immunoglobulins [2]. Thus, there is growing evidence that the role of B cells in MS extends well beyond the secretion of antibodies. Lately, B cell research in MS has shifted more towards their role in T cell and glial cell activation, cytokine secretion, and antigen presentation [15]. In the brains of MS patients, B cells have been found to accumulate, particularly in the perivascular and subarachnoid spaces of SPMS patients [16,17], which correlates well with local demyelination and neurodegeneration [18]. Hence, migration of B cells over the CNS barriers appears to be a crucial step in MS pathology. Studying the route of B cell entry into the brain and how the inflamed CNS milieu can sustain B cell survival is a crucial next step to better understanding the progression of this disease. In this review, we address the roles of different CNS barriers in MS pathology and how these barriers are involved in B cell migration. Moreover, we discuss how different MS therapies can impair B cell infiltration into the brain and how that relates to different clinical outcomes in MS.

## 2. Immune Cell Trafficking across the Different CNS Barriers

Limited numbers of peripheral immune cells can migrate into CNS compartments during homeostasis to act as sentinels in the surveillance of the CNS, whereas under neuroinflammatory conditions, a multitude of cells crosses the CNS barriers. Extravasation of cells from the blood into the tissue is a multi-step process. In general, the immune cell is apprehended from the bloodstream by selectins (e.g., E-selectin, P-selectin) located on endothelial cells, which interact with leukocyte glycoproteins, such as P-selectin glycoprotein ligand-1 (PSGL-1). This weak and transient interaction results in the tethering and rolling of the immune cell along the vessel wall [19]. Subsequent firm adhesion of the leukocyte to the inflamed endothelial cells halts the immune cell, which is mediated by cellular adhesion molecules (CAMs), such as immunoglobulin family members, cadherins, or integrins. For example, leukocytes express integrins, such as lymphocyte function-associated 1 (LFA-1) or very late activation antigen-4 (also known as α4β1 or VLA-4) that respectively bind to endothelial vascular cell adhesion molecule 1 (VCAM-1) and intercellular adhesion molecule 1 (ICAM-1) [19]. Other examples of endothelial CAMs are the melanoma cell adhesion molecule (MCAM) and activated leukocyte cell adhesion molecule (ALCAM) [20]. Chemokines are also essential regulators of the transendothelial migration of immune cells since they enhance the affinity of leukocyte integrins to bind strongly to endothelial CAMs [21,22]. Following this firm adhesion, leukocytes can cross the endothelium via paracellular or transcellular migration [23].

Three types of CNS barriers have been described through which immune cells can infiltrate the brain. A large bundle of research focused on the BBB, while fewer studies paid attention to the blood–meningeal barrier (BMB) and the blood–CSF barrier (BCSFB) in the choroid plexus (Figure 1). Together, these barriers protect the brain from peripheral damage and control the movement of molecules and cells from the periphery into the CNS. Although the three barriers share some features, their localization and anatomy can help understand how they differentially shape the CNS compartments.

### 2.1. Blood–Brain Barrier in Health and MS

The microvasculature of the brain parenchyma has several unique properties, creating a tightly regulated barrier known as the BBB. The BBB consists of unique continuous non-fenestrated cerebral endothelial cells (BECs) tightly connected by tight (TJ) and adherens junction (AJ) complexes. The barrier function of the BECs is further supported by the interaction with astrocyte endfeet, pericytes, neighboring microglia, and a continuous basement membrane. Altogether, this structure is called the neurovascular unit (Figure 1a) [24,25].

The junctional complexes TJ and AJ are connected to the cytoskeleton and can alter the morphology of the endothelium. These dynamic structures change depending on the local microenvironment and can also activate intracellular signaling pathways. Well-known TJ proteins of the brain endothelium are occludin, claudins, and junctional adhesion molecules (JAMs), while vascular endothelial cadherin (VE-cadherin) and platelet endothelial cell adhesion molecule 1 (PECAM-1 or CD31) are part of the AJs [24,25,26,27,28]. In addition, these complexes regulate the polarization of BECs by differentiating between apical and basal domains. Moreover, BECs express specific polarized transporters that allow the active transport of essential molecules and waste products [24,25,26,27,28]. Pericytes are contractile cells that regulate cerebral blood flow by interacting physically with BECs. In the neurovascular unit, pericytes are situated in between the BECs and astrocyte endfeet (Figure 1a) [29]. The astrocyte endfeet interact with the brain endothelium creating the glia limitans, which regulate the blood (and ion) flow and volume that passes through the capillaries [30,31]. Another component of this unit is the basement membrane. Astrocytes and BEC can secrete fibrous proteins or proteoglycans and generate this extracellular matrix, which maintains the structure of the neurovascular unit. In addition, it modulates BBB function and permeability since matrix proteins can influence the expression of TJs [32]. Finally, microglia are the CNS-resident immune cells involved in the homeostasis and protection of the CNS against pathogens or damage. Microglial processes can take over the coverage of pericytes or astrocytes on BECS and physically interact with the endothelium [33,34].

BBB dysfunction is one of the early key hallmarks of MS pathogenesis. Peripheral immune cells and CNS-resident cells (e.g., microglia or astrocytes) induce inflammation in BECs by secreting pro-inflammatory cytokines, such as tumor necrosis factor-α (TNF-α) or interferon-γ (INF-γ) [35]. This neuroinflammation associates with several molecular changes. (1) Inflamed BECs increase the presentation and secretion of chemokines and (2) enhance the expression of CAMs, which promote leukocyte transendothelial migration, also called diapedesis [20]. (3) This inflammatory state further modifies the structure and location of TJ and AJ, leading to a mesenchymal state and thereby increasing BBB permeability [36]. Consequently, leukocytes, such as B and T cells, cross the inflamed endothelium more easily. Immune cells can then stay in the perivascular spaces to create the perivascular immune aggregates characteristics of MS, but also later cross the glia limitans to infiltrate the brain parenchyma. Furthermore, in MS, these perivascular spaces are enlarged [37,38]. Once within the CNS, activated lymphocytes propagate a cascade of neuroinflammatory reactions leading to disease onset and progression and associated clinical symptoms [39,40].

### 2.2. Blood–CSF Barrier in Health and MS

The BCSFB is located in the choroid plexus in each of the brain ventricles. The choroid plexus is responsible for producing the CSF, supplying nutrients to the brain, clearing toxic molecules, and, thereby, maintaining brain homeostasis. It is a highly vascularized tissue and, consequently, a door for peripheral immune cell migration into the CSF. The architecture of the choroid plexus consists of a vascularized stroma surrounded by a layer of cuboidal epithelial cells (Figure 1b). The choroid plexus epithelium presents apical villi to increase the flux of solutes and water from the blood to the CSF. Moreover, apical motile cilia contribute to the CSF flow throughout the ventricular system [41]. The CSF contains compounds that help the CNS to develop and function normally, including water, ions, glucose, growth factors, amino acids, lipids, and hormones, among others [42]. Most of these components are imported from the blood or produced by the choroid plexus epithelium [41].

The choroid plexus stroma is abundantly populated by immune cells, mostly antigen-presenting cells [43,44], which guarantee immunosurveillance for the maintenance of a healthy brain. In contrast to most BECs, the choroid plexus endothelial cells are fenestrated and have higher constitutive expressions of ICAM-1 and P- and E-selectin [45,46] and lower amounts of TJ (Figure 1b) [47]. Hence, the choroid plexus capillaries allow easy immune trafficking to the choroid plexus stroma. Instead, the blood–CSF barrier is mainly formed by the choroid plexus epithelial cells. As a highly polarized barrier, the basal side of the epithelium interacts with the stroma, while the CSF-facing apical side is tightly connected with TJ, which hinders the migration of immune cells.

As a model system, the human cell line of choroid plexus epithelial cells (HIBCPP) is frequently used. While the HIBCPP cell line expresses ICAM-1, we found no evidence for the expressions of ICAM-1 or VCAM-1 in choroid plexus epithelial cells in healthy humans, only on the endothelial cells of the choroid plexus [43,45,48]. This suggests that other adhesion molecules might be involved in the transepithelial migration to guarantee immunosurveillance in healthy humans. Of note, the HIBCPP cell line comes from a human choroid plexus papilloma and the results obtained from this research model should be carefully considered [48]. Interestingly, both ICAM-1 and VCAM-1 are constitutively expressed on choroid plexus epithelial cells in mice, but not on the fenestrated endothelial cells [49]. Furthermore, the localization of ICAM-1 on the apical side of the epithelium suggests that it can be involved in cellular migration from the ventricular CSF to the stroma, as was shown in mice [50]. This process may be related to the reactivation of T cells and CSF monitoring and is thought to occur at a much lower rate than the infiltration into the CNS via the BBB [50,51]. Hence, the choroid plexus epithelium might allow a bi-directional migration of immune cells between blood and CSF in health.

In the early stages of MS, the choroid plexus has been described as an immunological niche for T cell activation in response to peripheral inflammation [51]. In EAE, pathogenic Th17 cells infiltrate through the choroid plexus via upregulation of CCR6 [52]. At the same time, choroid plexus epithelial cells in mice and humans express CCL20, a chemotactic signal for pathogenic Th17 cells [52]. Accordingly, human Th17 cells preferentially migrate through the cell line HIBCPP compared to other CD4^+^ T cell subsets [50]. Additionally, in EAE, there is increased expression of ICAM-1 in the epithelium of the choroid plexus [53]. Hence, it might be an important route of immune cell migration in MS. Furthermore, patients with MS had enlarged choroid plexi than the controls [54], which could be explained by the accumulation of leukocytes and/or edema [44].

In progressive phases of MS, the choroid plexus expresses low levels of the TJ claudin-3 compared to choroid plexus tissue from control donors [55], inflammation becomes chronic, and unknown triggers (such as hypoxia) may sustain the upregulation of adhesion molecules and chemokines [43,56]. Interestingly, B cells and plasma cells are virtually absent in the choroid plexus from control and MS donors, suggesting that the BCSFB is not the preferred route of entry for those cells in MS [43]. Furthermore, the choroid plexus may become an “educational gate” [44,57,58] and shift to selective recruitment of suppressive immune cells as seen with a specific accumulation of CD56^bright^ NK cells in MS [57].

### 2.3. Blood–Meningeal Barrier in Health and MS

The meninges consist of three layers of connective tissue that wrap the brain parenchyma: pia mater, arachnoid mater, and dura mater (Figure 1c) [58]. The structural role of the meninges is to structurally protect the CNS by anchoring the brain to the skull and preventing side-to-side movement of the brain and spinal cord injury [59]. The meninges also serve as an additional barrier to control the movement of molecules and cells between the periphery and the CNS [60].

The dura mater is the outer membrane located adjacent to the skull. It consists of two epithelial layers of dense collagen fibers [61,62]. The outermost layer is called the periosteal layer and is tightly adhered to the skull cap [61]. Internally attached to the periosteal layer is the meningeal layer. Nerves, arteries, veins, and lymphatic vessels run between the two dural layers similar to peripheral tissue [60]. There is no strict barrier between the blood vessels and the dura mater because dural vessels are fenestrated, thereby allowing the transport of small molecules to move the blood to the dura mater [61,63,64]. Furthermore, at the dural venous sinuses, there is a unique communication interface between the CNS and the immune system. While there is a low expression of TJ claudin-5 and occludin in the dural endothelial cells, the high expression of VCAM-1 and ICAM-1 in these cells together with the presence of dural lymphatics suggests that the dura mater might be an important site for immunological surveillance [63]. This is illustrated by the presence of different immune cell subtypes in the dural sinuses, including macrophages, dendritic cells, neutrophils, innate lymphoid cells, T cells, and B cells [64,65,66]. The subdural meninges comprise the arachnoid and pia mater, and in combination, they are often called leptomeninges as they are structurally connected [67]. Directly attached to the meningeal dural mater layer is the arachnoid mater, a translucent multilayer of dense leptomeningeal cells [62]. The outer layer of the arachnoid mater is joined by TJs and desmosomes and expresses efflux pumps forming an impermeable barrier for molecules and cells similar to the BBB [62,68]. In between the arachnoid mater and pia mater lies the subarachnoid space (SAS), which is filled with CSF that can be reabsorbed to the systemic circulation or lymph nodes through arachnoid granulations and villi and the meningeal-dural sinuses [69]. Thus, the arachnoid barrier controls the passage of CSF from the SAS into the dura mater and the entry of immune cells and molecules derived from dural arteries into the SAS [62]. Within the SAS, strands of collagen covered by a layer of leptomeningeal cells, called subarachnoid trabeculae, connect the inner layer of the arachnoid mater to the pia mater [67]. The leptomeningeal cells enclosing the subarachnoid trabeculae create a continuous cellular monolayer over the pia mater. This layer of leptomeningeal cells is connected by gap junctions, including connexins 26 and 43, which serve as semipermeable membranes for solutes [62,68,70]. The pia mater, the innermost layer of the meninges, houses blood vessels that penetrate the brain parenchyma and will form part of the BBB (Figure 1c).

Various vessels run through the leptomeninges and are suspended by the subarachnoid trabeculae [62]. To enter the SAS from a meningeal vessel, a cell or molecule has to cross the blood–meningeal barrier, which is composed of a layer of non-fenestrated endothelial cells connected by TJ and the pia mater [61,71,72]. While the meningeal EC characteristics remain in great part unknown, this barrier differs slightly from the features of BBB by lacking pericytes and the astrocytic endfeet [72,73]. Additionally, meningeal ECs have a higher constitutive expression of CAMs. In contrast to BECs, meningeal ECs express high levels of ICAM-1, even in a non-inflammatory environment [46], which may make vessels more permissive for immune cell transmigration in healthy states.

Initially, the meningeal barrier was thought to only provide physical protection to the CNS. However, it was recently shown that leptomeninges may be structures of immune cell reactivation before infiltrating the brain parenchyma under inflammation [16,18]. These lymphoid-like structures are called ectopic/tertiary lymphoid follicles that consist of aggregates of antigen-presenting cells (APC), T cells, and B cells. These structures are often present in chronic-progressive MS [16]. Autoimmunity is likely to be re-initiated in the meninges through the reactivation of reactive T cells by APCs in these ectopic lymphoid follicles. Progressive MS patients with these structures are characterized by a faster progression of the disease and earlier onset of neurological disability [74,75].

Under inflammatory conditions, lymphocytes and myeloid cells can easily cross the BMB as the meningeal blood vessels are permeable [61,73,76]. Hence, these vessels are primary entry sites for immune cells into the SAS [72]. In the animal model EAE, activated T cells adhere to the leptomeninges once they have migrated across leptomeningeal vessels. Non-activated T cells can be observed in the CSF, where they can exert immune surveillance or they are removed from the CNS through drainage of the CSF [77]. Previous studies in EAE showed how T cells and dendritic cells enter the leptomeningeal space before the onset of CNS inflammation [78]. This might suggest that migration across the BMB occurs earlier than over the BBB. This process could be mediated by P-selectin, which is upregulated before other endothelial adhesion molecules in the meninges and choroid plexus [79]. Moreover, meningeal inflammation also occurred in the early stages of MS before the emergence of white matter lesions [80], and it was frequently close to gray matter lesions, BBB damage, and cortical demyelination [75,80]. Thus, the entry of immune cells to the CNS cortex is likely to be preceded by infiltration of the meninges via the meningeal blood vessels.

The importance of the route of entry across the BMB has also been demonstrated with CXCR7 inhibitor-treated EAE animals. This inhibitor reduced leukocyte trafficking from the leptomeningeal vessels into the SAS and decreased the extent of parenchymal leukocyte infiltrates [81]. Furthermore, VCAM-1 is expressed under normal conditions in the human meninges and its expression is increased in the meninges of MS patients [82]. Altogether, these results indicate that the BMB represents an essential structure regulating the entry of immune cells to the CNS and that the meningeal compartment plays an active role in neuroinflammatory diseases, such as MS.

## 3. B Cell Migration across CNS Barriers

To date, B cells are now recognized as important cellular mediators in the pathogenesis of MS. Although the molecular pathways that mediate T cell migration are greatly understood, relatively little is known about how different B cell subsets gain access to the CNS. The different subsets and their route of migration are reviewed below.

### 3.1. B Cell Development and Function

In brief, B cells develop from hematopoietic stem cells in the bone marrow, initially as pro-B cells and then pre-B cells. B cells undergo immunoglobulin gene rearrangement to become immature B cells and mature in the periphery. Mature naïve B cells will encounter an antigen during an immune response and will become activated. They will differentiate into short-lived plasma cells or germinal center B cells. The germinal centers are also the sites in which B cells undergo immunoglobulin class switch recombination. After engaging with an antigen in the germinal centers, B cells will proliferate and eventually differentiate into long-lived memory and/or plasma cells (Figure 2) [83,84].

B cells are key mediators of adaptive immunity; their functions in MS have been reviewed extensively in the past years and are out of the scope of this review [15,85,86]. In essence, B cells are known for their antibody production. However, they are also antigen-presenting cells and can shape the responses of other immune cells. Furthermore, B cells produce both pro-inflammatory and anti-inflammatory cytokines and this secretion is altered in MS. Moreover, B cells are likely contributors to the formation of ectopic lymphoid follicles as seen in the meninges of SPMS patients [15,85,86].

### 3.2. B Cells in CNS Immunosurveillance

The first study, suggesting a role of B cells in the surveillance of healthy human brains, found low levels of these lymphocytes within the brain displaying an activated phenotype (CD20^+^ CD23^+^ cells). These B cells were sporadically observed in the parenchyma throughout the healthy brain and no perivascular B cells were found [87]. An extremely high ratio of 19,000:1 CD19^+^ cells was observed when comparing the B cell numbers in peripheral blood and CSF, respectively [45]. In mice, B cells are mostly abundant during brain development, contributing to oligodendrogenesis and myelination, but not so much in adulthood [88,89]. Interestingly, the migration of B cells across the BBB of healthy rats depends on the presence of antigens within the brain. Infusion of a T-dependent protein antigen using a brain parenchymal catheter led to the recruitment of B cells and antibody production, regardless of having an intact BBB [90]. This also showed that peripheral B cells can detect antigens behind a closed BBB and antigen-specific B cells can respond to this antigen within the brain. Thus, the detected small amount of B cells relative to T cells in the brain/CSF and the lack of migration without antigens challenge whether B cells actually infiltrate the CNS in health.

### 3.3. B Cell Locations in MS

In MS, B cells have been found in different CNS compartments, including in meninges, normal-appearing white matter, white matter lesions, the cortex, and the CSF [91]. Within the CNS and the CSF, B cells share properties of a post-germinal center, antigen-experienced, and class-switched B cells [92,93]. Indeed, analysis of the antibody and B cell receptor repertoire indicated that several clonally expanded B cells overlapped between the blood, CSF, meningeal aggregates, and parenchyma infiltrates, although many clones were exclusive to each compartment [94,95,96,97].

Clonal B cells in the periphery and the different brain compartments in MS are evidence that B cells can traffic across the different CNS barriers. However, little is known about the mechanisms of B cell migration across these barriers and how this trafficking depends on the expressions of different CAMs or chemokines by BECs, meningeal endothelial cells, and choroid plexus epithelial cells; and whether different B cell subsets might have a preferential barrier to migrate across.

### 3.4. B Cell Migration across BBB

In the first studies on B cell migration, B cells and CD8^+^ T cells seemed to adhere more efficiently to rat brain microvascular endothelial cells than CD4^+^ T cells while CD4^+^ T cells migrate faster across endothelial cells [98,99]. However, contradictory results were found where B cells migrated more efficiently across human brain endothelial cells than CD3^+^ T cells from the same individuals [100]. These experiments clearly show that each immune cell type might have a different selective adhesion capacity to brain endothelial cells and further research is required to elucidate the migratory capacity of different lymphocytes.

Endothelial cells of the BBB in MS lesions upregulate VCAM-1 and ICAM-1 at the sites of B cell infiltration. Of note, VCAM-1 seemed to be higher-expressed in chronic MS lesions while ICAM-1 was present more uniformly in all lesion types [101]. In addition, circulating B cells constitutively express their respective counterreceptors VLA-4 and LFA-1 [100,101]. This suggested that B cells might use these CAMs to migrate into the CNS (Figure 3).

Blocking VLA-4 on B cells reduced B cell migration across human BECs, whereas antibodies against VCAM-1 did not inhibit migration [100,102]. This suggested that VLA-4/VCAM-1 interaction might not be necessary for transendothelial B cell migration and that VLA-4 might be using another endothelial ligand to allow B cell migration. Indeed, studies indicate that this ligand was fibronectin, an extracellular matrix protein involved in BBB integrity [100]. Blocking ICAM-1 also reduced B cell migration in vitro, although the effect was less pronounced as seen with VLA-4 blockage [100,102].

In an animal model of EAE, in which both B and T cells were required for pathology, specific deletion of VLA-4 on B cells reduced the recruitment of B cells, Th17 cells, and macrophages into the CNS, which resulted in a modest but significant reduction of the clinical symptoms. VLA-4 deletion did not modify peripheral B cell or T cell activation, indicating that the results were due to the selective blockage of B cells into the CNS [103]. As expected, these results were not replicated in a T-cell dependent EAE model after α4-integrin deletion in B cells, which together with integrin β1 forms VLA-4. In this model, B cell infiltration into the CNS was reduced but the clinical symptoms did not improve [104]. In a similar experimental set-up with α4-integrin deletion, more severe EAE was associated with the absence of regulatory B cells (Breg) in the CNS, indicating that VLA-4 was required for Breg infiltration into the brain [105]. Recent research shows that dopamine signaling increased the expression of α4-integrin in vitro on mice CD21^+^ CD23^+^ IgM^+^ Bregs cells [106]. These data confirm that VLA-4 is needed for Breg tropism in the CNS. Taken together, there are controversial data in mice regarding the effect of VLA-4 on B cell migration and positive outcomes, but it is clear that VLA-4 is important for B cells to cross brain endothelial cells in vitro and that its blockage is beneficial in MS, as seen with natalizumab, an anti-VLA-4 blocking antibody therapy.

Another molecule that has been studied in B cell migration is L-selectin, which is involved in the rolling step before crossing the endothelial cells. When depleting L-selectin in EAE, perivascular infiltrates with B cells were still found in the CNS, suggesting that L-selectin is likely not involved in transendothelial B cell migration in EAE [107].

Recent work shows the involvement of ALCAM in B cell migration in the CNS. ALCAM was initially described as a co-stimulatory molecule forming immune synapses between T cells and antigen-presenting cells [108]. Now we know that around 50% of CD19^+^ B cells express ALCAM and that activated B cells in MS upregulate this molecule [102]. Interestingly, both ALCAM^+^ and ALCAM^-^ B cells in mice co-express other CAMs, such as ICAM-1, VLA-4, and CD11a, but in humans, only ALCAM^+^ B cells co-express them [102]. This again pinpoints differences between mice and humans. Reduced disease severity was observed upon ALCAM blockage, which suggested a role of ALCAM in immune cell transmigration. Indeed, antibodies against ALCAM reduced the adherence of CD19^+^ B cells to human brain endothelial cells and their migration in vitro by around 40%. Thus, blocking ALCAM was not enough to completely prevent B cell migration, suggesting that other molecules are required in this process [102].

Regarding B cell subsets, memory B cells adhere more effectively to ICAM-1 and VCAM-1 compared to naïve B cells in health [109]. In MS lesions, mostly switched memory B cells are present in the perivascular space, but also a perivascular accumulation of naïve B cells has been observed [110]. Switched memory B cells have also been shown to infiltrate further into the parenchyma of active lesions [110,111], although this was observed in a case with a severe rebound after natalizumab treatment. The fact that switched memory B cells are found further away from vessels might indicate that these cells may have migrated first or that naïve B cells could differentiate into switched memory B cells within the CNS parenchyma [96,112]. Interestingly, plasma cell numbers increased in active lesions after blockage of VLA-4 in natalizumab-treated patients [113]. Furthermore, natalizumab treatment reduced B cell migration but did not stop it completely [114]. Thus, blocking VLA-4 did not completely prevent the infiltration of immune cells in the brain. It would be interesting to study if plasma cells are not affected by VLA-4 blockage or if specific subsets of B cells can migrate across the BBB and differentiate into plasma cells in the lesion.

Overall, data show that more than one cellular adhesion molecule is needed for B cell recruitment into the CNS parenchyma; that there are marked differences in this process between mouse and human data, and that different B cell subsets might use different mechanisms to cross the BBB.

### 3.5. B Cell Migration across BCSFB

Immune cell entry into the CNS via the choroid plexus requires the initial infiltration from the peripheral circulation in the choroid plexus stroma, and the second step of transepithelial diapedesis from the choroid plexus stroma into the CSF. The choroid plexus contribution to immune cell migration into the CNS might be lower than that of the BBB [50]. This is in part because the choroid plexus epithelium might form a tighter barrier than the BBB [50]. However, certain immune cell populations may be more prone to migrate through the BCSFB relative to the BBB, and CSF chemokines can act as additional cues to attract immune cells into the CNS. Furthermore, we need to be aware of the differences observed in in vitro models compared with human tissue. In MS brains, the choroid plexus does not appear to be an important niche for B cells and/or plasma cells since they are rarely observed there [43,44].

A recent paper showed that unstimulated CD19^+^ B cells migrate scarcely across the human cell line HIBCPP and that this process required various chemokines, such as CXCL12 and/or CXCL13 [115]. These chemokines are known to regulate B cell migration into lymphoid tissue and also attract B cells to inflamed CNS sites [16,21]. Interestingly, MS patients have increased levels of CXCL12 and CXCL13 in the CSF, and B cell and plasmablasts numbers in the CSF correlate well with the concentration of CXCL13 in the CSF [74,116]. After migration, CD19^+^ B cells increased the expression of chemokines, but no differences were found in LFA-1 and VLA-4 before and after migration [115]. It would be interesting to block ICAM-1 and/or VCAM-1 in the HIBCPP cell line, to confirm their role in B cell migration across this model. CD27^+^IgD^-^ class-switched memory cells from MS donors crossed the cell line HIBCPP in higher numbers than controls. However, migrated B cells were still in low numbers and no differences were observed between MS patients with active disease or clinical remission [115].

We recently showed (with a high dimensional single-cell approach) that while B cells are increased in the blood, the percentage of B cells out of the CD45^+^ immune population is very low in the choroid plexus of MS donors [57]. Of note, we could observe a slightly higher percentage of T-bet^+^ B cells in the choroid plexus of the control brains, although it was non-significant [57]. This could be explained by an increase of age-associated B cells, which are known to express T-bet^+^, as similarly seen in the brains of aged mice [89,117].

Different subsets of B cells increase in the CSF of MS patients, including class-switched memory B cells, short-lived plasmablasts, and some possible germinal center B cells [112,118,119]. These subsets were not found in the choroid plexus of progressive MS donors. This could indicate that the choroid plexus might not be an important route for B cell recruitment and that other routes, such as the meninges, might be better niches for B cell infiltration and persistence in this phase of the disease (Figure 3). This can be supported by previous research showing how clonally-expanded B cells from the meningeal compartment of progressive MS can be detected in the CSF and parenchymal infiltrates [94]. Of note, we cannot rule out the fact that the choroid plexus might be an important barrier in relapsing-remitting MS but further research is needed to confirm if that is the case.

### 3.6. B Cell Migration across BMB

The meninges seem to be an important location in the pathogenesis of MS as immune cells might enter the meninges even before infiltrating the brain parenchyma and auto-immunity is likely to be reinitiated in the meninges through reactivation of reactive T cells by APCs in follicle-like structures [16,18,78,79]. Since the meninges are the primary collection site for B cells in MS and the number of B cells increases during MS progression and associates with pathology [17], their migration across the BMB is a crucial process, but the mechanism remains largely unknown.

Overall, B cells migrate more effectively across meningeal endothelial cells than BECs [102]. This suggests that the endothelial cells of the meninges allow for more B cell migration than endothelial cells from the BBB. Different integrins have been hypothesized to be involved in this process, such as VLA-4 and LFA-1 since T cells use them to attach to the leptomeninges after egressing from the leptomeningeal vessels [77]. The importance of VLA-4 has also been highlighted by a study showing that VLA-4 is involved in the accumulation of B cells in the meninges during neuroinflammation in EAE. Neutrophils, which traffic into the CNS first via the chemokine receptor CXCR2, coordinate B cell recruitment into the meninges in a VLA-4-dependent manner [120]. In addition, blocking antibodies against ALCAM, ICAM-1, or VCAM-1 in primary human meningeal endothelial cells reduced B cell migration by around 20%, 40%, and 60%, respectively. This confirms that these three molecules are all involved in B cell recruitment into the leptomeninges but the interaction between VCAM-1 and VLA-4 might be the most important one for B cell migration (Figure 3) [102]. In meningeal follicle-like structures, B cells can exhibit germinal center activity [121] and interact with follicular dendritic cells to become memory B cells. Interestingly, this interaction involves the adhesion of VLA-4 and LFA-1 with the respective ligands ICAM-1 and VCAM-1 on dendritic cells [122]. Therefore, these cell adhesion molecules are not only involved in adhesion and migration but also the activation and differentiation of B cells. It would be intriguing to study if B cell interaction with meningeal endothelial cells provides the same activation and differentiation of B cells.

In paired blood and CSF MS samples, CD19^+^IgD^+^CD27^−^ naïve B cells were more abundant in the blood while CD19^+^IgD^+^CD27^+^ memory B cells accumulated in the CSF. Interestingly, CD19^+^CD38^high^CD77^+^ centroblasts—a population present in secondary lymphoid organs—were only found in the CSF of MS patients [112]. Centroblasts from ectopic lymphoid follicles may differentiate from circulating naïve B cells and/or from memory B cells that have migrated into the CNS [123]. It is tempting to speculate that since there are fewer naïve B cells in the CSF, these CSF centroblasts might derive from migrated memory B cells. Thus, memory B cells may migrate into the meninges and generate the appropriate environment for B cell differentiation into centroblasts, forming these ectopic lymphoid follicles. Using both bulk and single RNA sequencing in paired blood and CSF MS samples, it was shown that naïve blood-derived B cells, in comparison with the CSF, express higher levels of ITGAM. ITGAM codes for the Mac-1 gene and Mac-1 is known to be involved in immune cell migration [124,125]. In contrast, unswitched memory B cells, plasmablasts, and plasma cells decreased the expression of the ITGA4 gene—α4 integrin—in CSF compared to blood [124,126]. This suggests that these cells in the blood use VLA-4 to migrate into the CSF, maybe via the BMB, and once there, its expression is lost. Targeting VLA-4 might not affect plasma cell migration into the CNS via the BBB [113], yet it would be interesting to study if this also holds for plasma cell migration across the BMB.

In the experimental animal model of opticospinal encephalomyelitis co-expressing MOG-specific TCR and BCR, ablation of ITGA4 on B cells reduced meningeal B cell numbers. Ectopic lymphoid follicles were still present but showed a decreased volume due to reduced numbers of B cells. Interestingly, after two months, the disease burden and mortality were higher in these animals compared to controls without ablation of ITGA4. This could be explained by the fact that one-third of the B cells in ectopic lymphoid follicles produce the anti-inflammatory cytokines IL-10 and IL-35 [127]. In contrast, in EAE mice with a selective B cell VLA-4 deficiency, the decrease of meningeal B cells also reduced EAE susceptibility. Other effector leukocytes, such as infiltrating macrophages and T cells—especially Th17 cells—were also reduced in the meninges [103]. Notably, in EAE, Th17 cells within the meningeal compartment express the transcription factor Bcl6, which is involved in the production of B cells supporting cytokines. This might create an ideal microenvironment for B cell tropism, differentiation of B cells into follicular B cells, and immunoglobulin class switching, thereby controlling meningeal lymphoid tissue formation [128]. In sum, meningeal B cells comprise a heterogeneous cell population with both cytotoxic and immune regulatory functions. Further research is needed to gain more insight into the functional role of meningeal B cells and to what extent they differentiate into various subsets in the meningeal compartment

The chemokines CXCL12 and, especially, CXCL13, are involved in the recruitment of B cells into the CNS and play a central role in the formation of lymphoid follicle-like structures in the leptomeninges [16,21,116]. Since CXCL13 is strongly associated with intrathecal immunoglobulin synthesis and composition of cells in MS lesions, it has been suggested to use CXCL13 levels in CSF as a predictor for B cell depletion therapy outcome [129]. However, in CXCL13-deficient EAE mice, the disease course neither changed nor was there any effect on B cell infiltration to the spinal cords. This suggested that CXCL13 is dispensable for B cell migration in this experimental model or at least for B cell migration in the spinal cord [130]. This may be plausible as many other chemokines might be involved in the recruitment of B cells into the CNS. For instance, CXCL10 attracts both T cells and antibody-secreting cells (ASC) to inflamed CNS sites [131]. Moreover, human BECs constitutively secrete CCL2, and B cells contain mRNA transcripts of the CCL2 receptors and antibodies targeted against CCL2 selectively inhibited transendothelial B cell migration [100]. Hence, more research needs to be performed to see which chemokines are primarily responsible for B cell recruitment into the meninges during MS.

In MS, B cells can migrate across the BMB to reside in the leptomeninges, but recently it was shown that B cells infiltrate from the skull bone marrow into the dura of healthy mice [89]. The calvaria is a hematopoietic region located in the cranial bones, which contains specialized vascular channels [132] that allow B cells to migrate from the bone marrow to the dura to mature locally [89]. This infiltration and B cell development might be facilitated via the interaction of the receptor CXCR4 on B cells and the expression of CXCL12 from dura fibroblast-like cells [89]. In there, autoreactive immature B cells against CNS antigens, such as MOG, are negatively selected and are thereby specifically eliminated from the meninges. As such, meninges can behave as a regulatory reservoir to eliminate autoreactive B cells and allow the correct homeostatic B cell development [133,134]. Interestingly, in healthy young mice, dural B cells did not clonally overlap with circulating B cells, suggesting that most meningeal B cells originated from the calvarial bone marrow. However, in healthy aged mice, 30% of the dural B cells were clonally present in the blood, indicating that these B cells have migrated from the periphery [89]. Whether meningeal B cell migration from both blood and calvarial bone marrow occurs in progressive MS is still unknown. Further studies are needed to understand how autoreactive B cells against CNS antigens are surviving in the meninges and if they preferentially migrate from the periphery or skull bone marrow. At the same time, B cells can exit the leptomeninges via dural lymphatic vessels and drain towards the cervical lymph nodes [89], where they can encounter antigens and undergo affinity maturation [95,135]. Interestingly, the presence of clonally expanding B cells in the CNS and draining cervical lymph nodes could suggest a bidirectional migration between these compartments [95]. However, the (molecular) mechanisms that drive this migration from the leptomeninges toward the lymphatic vessels are currently unknown.

In conclusion, recent research has shown that ALCAM, ICAM-1, and VLA-4 are involved in the transmigration of B cells across the human BMB and in animal models of MS. VLA-4 seems to be an essential integrin for B cell migration across the meningeal endothelial cells with the help of different chemokines. However, as the meninges are an ideal environmental niche for B cells in MS, it is difficult to speculate how different B cell subsets might employ different migration mechanisms. It is also unknown how or if the first autoreactive B cell arrived in the leptomeninges during health from the skull bone marrow or during the disease from the periphery.

## 4. Effects of MS Therapies on B Cell Migration

Knowing that the localization of B cells and the molecular players involved in B cell migration into the CNS is important from a therapeutical point of view, we now focus on how several MS therapeutics that are approved or in late-stage development affect B cell migration.

### 4.1. Sphingosine 1-Phosphate Receptor Agonists or Modulators

Fingolimod is a sphingosine 1-phosphate (S1P) receptor agonist that blocks lymphocyte egress from the lymph nodes into the circulation. Different cells, such as lymphocytes and neural cells, express S1P receptors. There are five different receptors and fingolimod lacks activity on one of them (S1P_2_). S1P is present in a gradient of low concentrations in the lymph nodes and high concentrations in blood, thereby attracting cells that express its receptor into the blood (mechanisms reviewed elsewhere [136]). Fingolimod binds with high affinity to S1P receptors, particularly S1P_1_ receptors, inducing their internalization and making them unresponsive to S1P. Consequently, fingolimod potentially sequesters encephalitogenic lymphocytes in the lymph nodes [137].

Fingolimod decreases the ratio of peripheral non-class-switched and class-switched memory B cells to naïve cells. Furthermore, it increases peripheral transitional, regulatory B cell subsets, as well as plasma cells [114,138,139,140]. Interestingly, this drug decreases ICAM-1^+^ B cells, which could indicate reduced antigen presentation capabilities and/or reduced migration across the CNS barriers [138]. However, regulatory B cells from MS patients treated with fingolimod displayed increased migratory capacity across a human BEC cell line and this was accompanied by an increased ratio of regulatory B cells in the CSF of MS patients. [141]. This could either suggest that fingolimod is involved in regulatory B cell differentiation or that S1P receptors are not important for preventing the migration of this subset. Although without statistical significance, fingolimod-treated B cells tended to migrate in fewer numbers across the cell line HIBCPP than untreated B cells [115]. Remarkably, the elevated frequency of clonal B cell groups between blood and CSF in fingolimod-treated patients suggests that B cell migration might not be affected [114]. Furthermore, fingolimod treatment has little effect on the number of B cells in the CSF [142]. Treatment of fingolimod at the beginning of EAE reduced the number of B cell infiltrates in the parenchyma, but in chronic EAE, fingolimod did not affect the maintenance of B cell aggregates in perivascular areas from the cerebellum [143]. Whether fingolimod affects the formation of ectopic lymphoid follicles in the meninges is still unknown.

Siponimod is a novel next-generation S1P receptor modulator that might be useful for the treatment of the progressive phases of MS [144]. It particularly binds to S1P_1_ and S1P_5_ receptors [145]. This small molecule can also cross the BBB and reach the CNS parenchyma as well as the meninges, thereby it could block lymphocyte migration to the meningeal compartment. Siponimod reduced EAE severity and halted the formation of meningeal ectopic lymphoid follicles [146]. It is still under debate if the reduction of this meningeal compartmentalized inflammation is enough to reduce the clinical symptoms of progressive MS, or if in the case of siponimod, other secondary effects have roles. For instance, with S1P receptor modulators, T and B cells shift toward a more regulatory phenotype [147], which could act both in the periphery but also centrally by inhibiting meningeal inflammation.

### 4.2. Cladribine

Cladribine (1-chlorodeoxyadenosine, 2-CdA) is a purine adenosine analog that is activated intracellularly when phosphorylated by deoxycytidine kinase (DCK). The triphosphorylated form disrupts DNA synthesis and repair, thereby inducing cellular death. In most cells, this form is inactivated via dephosphorylation with the phosphatase 5′-nucleotidase. However in lymphocytes, especially B cells, the ratio kinase to phosphatase is high and, consequently, they accumulate cladribine phosphates. Consequently, cladribine preferentially depletes the lymphocyte populations (mechanism reviewed at [148]).

Cladribine quickly eliminates peripheral B cells, especially class-switched and unswitched memory B cells [149,150]. Notably, germinal center B cells and memory B cells exhibit high levels of DCK [149], which suggests that they are potentially vulnerable to cladribine. This can lead to long-term loss of memory B cells in the blood since they repopulate slowly from lymphoid tissues [151]. It would be interesting to study if meningeal B cells also have higher DCK activity and if they could be specifically targeted. Untreated MS patients have a higher percentage of ICAM-1^+^ CD19^+^, LFA-1^+^ CD19^+^, and PSGL-1^+^ CD19^+^ B cells. Notably, after two years of treatment with cladribine, these B cell populations declined to similar values as healthy controls [151]. This suggests that cladribine not only quickly depletes memory B cells, but that in the long run, it might have an effect on B cell migration across the CNS barriers.

### 4.3. Natalizumab

Natalizumab is a monoclonal antibody that targets CD49d—the subunit α4 of VLA-4—thereby blocking its binding to VCAM-1 on endothelial cells (reviewed in [152]).

Natalizumab treatment elevated the frequencies of different B cell subsets, such as pre-B cells, mature B cells, transitional B cells, and memory B cells in the blood [114,115,153,154,155,156]. However, natalizumab also decreased peripheral plasmablast frequency in both progressive MS and RRMS. Interestingly, in healthy donors, plasmablasts have a higher expression of CD49d in comparison with other B cell subsets [154]. In one study, natalizumab-treated MS patients did not contain plasmablasts in the CSF [154], whereas, in another study, natalizumab treatment resulted in increased plasmablasts and plasma cell numbers in the CSF [113]. Furthermore, plasma cells are still present in brain lesions after natalizumab treatment, suggesting that other CAMs might be involved in the migration of plasma cells or that recently migrated B cells can persist locally and differentiate into plasmablasts and plasma cells regardless of natalizumab treatment.

Progressive multifocal leukoencephalopathy (PML) is a rare demyelinating disease caused by John Cunningham polyomavirus (JCV). This infection commonly occurs in childhood and afterward, the virus remains latent, often in lymphoid organs [157]. Immunosuppressive treatments for MS, such as natalizumab, can lead to either reactivation of the virus or facilitating the mobilization of the virus into the CNS, which leads to acute and severe demyelination. Interestingly, the occurrence of PML may be related to B cell trafficking as B cells may become a viral reservoir and candidates to mobilize the virus in the CNS [158,159]. It has been suggested that blocking α4 integrin might increase the number of peripheral JCV-infected B cells coming from the bone marrow, thus, facilitating PML development [160].

Targeting VLA-4 has been very useful to study immune cell migration in vitro and in animal experiments as shown in previous sections of this review. All the cited literature shows how VLA-4 is an important factor involved in B cell migration across different CNS barriers [100,102]. Current studies of natalizumab now also suggest that different B cell subsets might use different molecular pathways to cross these CNS barriers.

### 4.4. Ocrelizumab

Ocrelizumab is a novel humanized anti-CD20 monoclonal antibody approved for treating RRMS and PPMS patients. By binding to CD20, ocrelizumab depletes B cells by complement-dependent lysis and/or antibody-dependent cytotoxicity [161].

Within weeks, ocrelizumab depletes almost all CD19^+^ B cells, especially naïve and memory B cells, except pro-B cells, and antibody-secreting cells, such as plasmablasts [162,163]. Ocrelizumab also removes peripheral CD20^+^ T cells [164], which are T cells that have acquired CD20 via trogocytosis while interacting with antigen-presenting B cells [165]. Some have speculated that depletion of CD20^+^ T cells explains the success of anti-CD20 therapies in MS, but this is out of the scope of this review [166,167]. Interestingly, B cells are also reduced in the CSF after ocrelizumab treatment [168], which can indicate that there is reduced B cell migration across CNS barriers or that ocrelizumab can deplete B cells within the CNS.

The animal model 2D2xTh leads to spontaneous development of EAE with meningeal lymphoid follicles but does not require B cells for pathology. Depleting peripheral B cells before the EAE onset did not halt the production of these follicles nor reduce clinical severity [169]. Depleting peripheral B cells after the onset of EAE also did not affect the number or size of the meningeal follicles. However, in-depth studies showed that one-third of the meningeal B cells were reduced [169]. The authors could not evaluate whether this depletion was due to a reduction of B cell trafficking or whether the antibody could go inside the meninges and start depleting B cells.

The opticospinal encephalomyelitis mouse model also develops spontaneous EAE and both T and B cells express MOG-specific antigen receptors [127]. Treatment with anti-CD20 antibodies at the pre-symptomatic phase reduced disease severity [127], which, in comparison with the animal model 2D2xTh, suggests that B cells might have antigen-presenting functions before the onset of the disease. Interestingly, anti-CD20 antibodies did not abrogate the formation of meningeal lymphoid follicles, although they were smaller than the ones in untreated animals. B cell depletion after the onset of symptoms neither modulated disease progression nor reduced the area of meningeal lymphoid follicles [127], indicating that meningeal-resident B cells might be resistant to anti-CD20 therapies. Hence, once B cells have trafficked into the intrathecal compartment—leptomeningeal or CNS parenchymal—anti-CD20 therapies might be less effective. Furthermore, targeting all CD20^+^ B cells has collateral effects since regulatory B cells can normalize EAE and their depletion is pathogenic for EAE [127,170]. Further research is needed to complete our understanding of the effects of anti-CD20 antibodies on B cell migration and survival behind the CNS barriers, and for this, new suitable animal models are needed. The chronic meningeal inflammation (CMI) animal model could be a suitable option to study in detail B cell migration in the meninges and the effect of anti-CD20 therapies [17,171].

### 4.5. BTK Inhibitors

Bruton tyrosine kinase (BTK) translocates from the cytosol to the cell membrane when an antigen binds to the B cell receptor (BCR). Afterward, BTK activates a pathway involved in BCR signaling and differentiation and survival of B cells (reviewed in [172]). BTK inhibitors (BTKi) are small molecules that have been recently researched for the treatment of MS [173]. Interestingly, BTKi can also cross the BBB [174].

Currently, the effect of BTKi on the phenotype of peripheral B cells in MS patients remains to be studied. BTKi might differentially modify various B cell subsets, such as memory B cells, which are important for the disease. In EAE, BTKi did not reduce the frequency of B cells in the blood, spleen, or lymph nodes, but it ameliorated EAE disease severity and reduced B cell and T cell infiltrates in the brain parenchyma [175]. Most importantly, only follicular B cells were reduced in numbers in the blood, spleen, and lymph nodes, and reduced the expression of activation markers involved in antigen presentation. It would be interesting to study if CAMs might also be decreased since BTKi inhibited B cell activation [175]. Since BTKi seems to particularly affect follicular B cells and their activation, further research is needed to study if BTKi affects the formation of meningeal lymphoid follicles. If so, combining, in time, anti-CD20 therapies with BTKi could provide benefits by targeting B cell trafficking into the CNS, and once they are behind these barriers, targeting their function.

## 5. Conclusions

Some of the most effective MS drugs are either depleting peripheral B cells—especially memory B cells—or blocking their transmigration into the CNS. Therefore, B cells have been in the spotlight of MS research for the past decade. While most studies have addressed the different functions of B cells in health and disease, it is surprising to note the virtual absence of CNS B cells in healthy states and specific compartments in the brains of MS patients, such as the choroid plexus. This is of special importance when compared with other immune cell subsets, such as T cells, known for their immunological surveillance and presence in almost every compartment of the CNS.

For a long time, the migrational mechanisms of B cells have been extrapolated from the vast knowledge of T cell trafficking into the brain via the BBB. However, knowing the different brain locations of these immune cells and the complexities of the immune system, and the different CNS barriers, it is easy to speculate that B and T cells use different migrational pathways for each CNS barrier. Furthermore, in this review, we have pinpointed how these CNS barriers differ in health and MS. We are convinced that a better understanding of the migratory process of B cells into the CNS would ultimately aid in the development of more refined therapies for MS and eventually stop its progression.

## Figures and Tables

**Figure 1 biomolecules-12-00800-f001:**
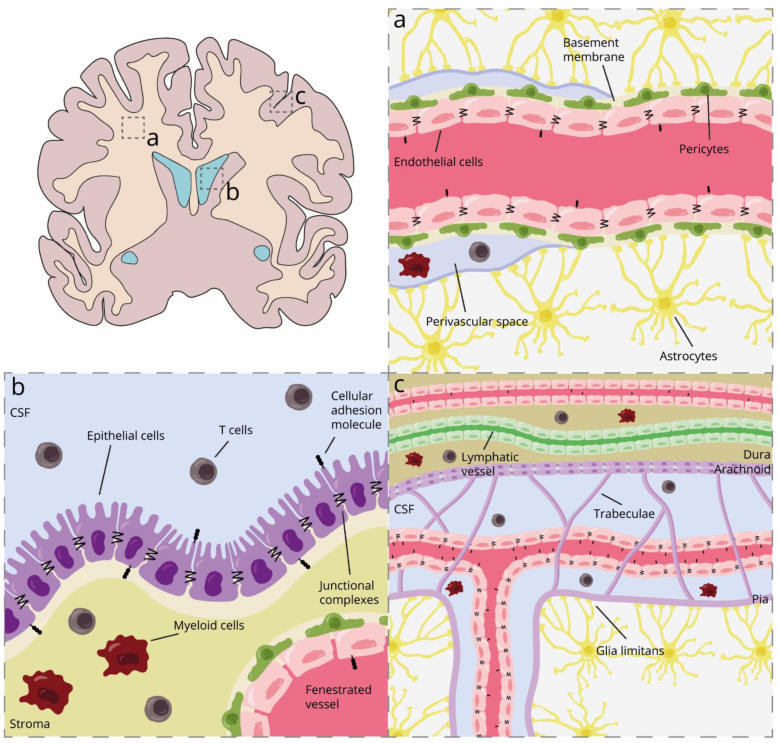
Schematic representation of the different CNS barriers and their localization in health. (**a**) Blood–brain barrier; (**b**) blood–CSF barrier in the choroid plexus; and (**c**) blood–meningeal barrier. CSF, cerebrospinal fluid.

**Figure 2 biomolecules-12-00800-f002:**
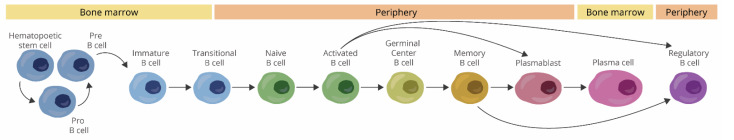
B cell development and differentiation.

**Figure 3 biomolecules-12-00800-f003:**
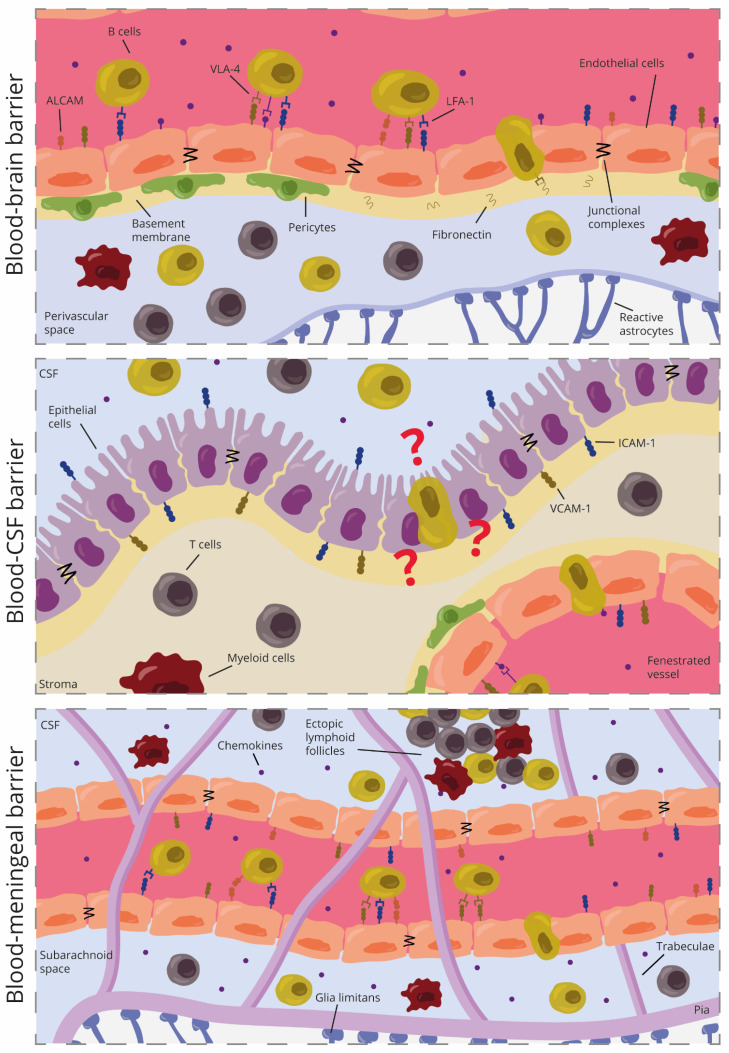
Schematic representation of B cell migration across the different CNS barriers in multiple sclerosis.

## Data Availability

All data and materials supporting the results of the present study are available in the published article.

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
