# Peer review of "Breaching Brain Barriers: B Cell Migration in Multiple Sclerosis"

_biomolecules, 2022, doi:10.3390/biom12060800_

Round 1
Reviewer 1 Report
Authors propose this review because they think that knowledge of the different routes of how B cells enter the inflamed CNS may be crucial to the understanding of MS pathogenesis. They present a review of studies which have addressed the different functions of B cells in health and pathological conditions as regards to CNS; they come up with the conclusion that B cells are absent in healthy CNS states and in specific compartments in the brains of MS patients, such as the choroid plexus.
This review is extremely important for the understanding of the pathogenesis of MS. Infact, researchers have, up to now, focused on the view that migrational mechanisms of B cells could be extrapolated from the vast knowledge of T cell trafficking into the brain via the BBB. This review contributes, on the contrary, to shade light on the different brain locations of B and T cells, which togegher with the complexity of the immune system and of the different CNS barriers, make it possible to speculate that B and T cells use different migrational pathways for each CNS barrier.
The review topic is extensively covered and references are appropriate.
Besides Authors examine mechanisms of action of Disease modifying Drugs (DMDs) used in MS, adding experimental data in terms of B cell trafficking, which may contribute to a deeper understanding of their efficacy and side-effects.
Suggestions to improve the review:
1) Authors may add some data on the different susceptibility to develop PML (Progressive Multifocal Leucoencephalopathy) in MS patients treated with natalizumab and ocrelizumab. Some data are arguing that its occurrence may be related to B cell trafficking.
2) English language may be improved.
Reviewer 2 Report
In this review, the authors summarize current knowledge on B cells in MS and especially focus the blood-brain barrier, Blood-CSF barrier, blood –meningeal barrier and B cell migration into the CNS. The review is well written and provides a very nice and comprehensive overview on the different barriers and the molecular mechanismns of B cell migration between the CNS and peripheral compartment. In addition, the effects of different MS treatments is outlined.
Major points:
- In general, a very detailed overview on B cell migration into the CNS compartment is provided, however, the efflux of B cells out of the CNS (drainage in cervical lymph nodes) is also extremely important considering antigen presenting properties of B cells in the periphery. This aspect should be also discussed in the manuscript.
- The approach for literature research is not outlined. It is extremly difficult to cite alls relevant MS papers in these kind of reviews. However, at some passages, the literature seem to lack some important papers (especially studies on B cell migration using NGS Ig repertoire sequencing).
- In e.g. chapter 2 I suggest to add a small paragraph on draining of CSF from subarachnoid space through channels in the cribriform plate into cervical lymph nodes.
- Page 8 line 351: The papers from the group von Büdingen 2012 / Palanichamy 2014 on this topic could be cited.
- Page 8, line 352: It is not well established whether B cells are restricted to specific brain compartments 352 or if they can traffic freely between them. -> I would not really agree with this sentence. It is well established, that B cells trafficking between the CSF compartment and the periphery. For review Ruschil et al., 2021 gives an overview on CSF B cell repertoires and their connection with the peripheral blood. Please reconsider this paragraph.
- Page 10: Regarding natalizumab and fingolimod, Kowarik et al., 2021 performed a study on B cell migration under both therapies by B cell receptor mass sequencing which should be cited in this context and at the later paragraphs.
- Page 14, line576: maybe the Stern et al. paper could be considered more in this paragraph.
- Page 14, line 581: However, as the meninges are an 581 ideal environmental niche for B cells in MS, it is difficult to speculate how different B cells 582 subsets might employ different migration echanisms. -> There is some data on the migration of different B cell subtypes across the BBB. Please reconsider this statement which is also find at previous paagraphs and refer to Palanichamy et al., 2014 and Kowarik et al., 2021.
- Paragraph 4.1. Fingolimod: Kowarik et al., 2021 (Neurotherapeutics) studied the effects of fingolimod on B cell trafficking in MS patients, please also refer to this paper (possibly also see Kowarik et al., 2011 Neurology).
- Paragraph 4.3. please also see paper mentioned above.
Minor points
- Page 1: After 10-20 years, approximately 80% of RRMS cases develop SPMS. This seems to be a quiet high number but might be different in the literature, please re-check.
- Page 13 line 520: blank space error.
Reviewer 3 Report
This article is an extensive review of the B-cell literature in the brain with an emphasis on the barrier entry points of the CNS and how B-cells might migrate in MS. The reviewer also speaks to how B-cells are modulated in current MS treatments – this part of the review would benefit from a table to summarize the treatments/effects on B-cells. It would also be interesting to discuss what we have learnt from failed B-cell therapies trials – are there instances where plasmablast targeting in fact makes disease worse?
Please see below for minor comments/grammar issues to address:
Fig 1 – be consistent with presence of immune cells across panels
Line 141 – glial limitans also extends beyond the capillaries
176 – expand on general role/s of epithelial cells as major secretory cells of CSF
180 – constitutive
210 – typo
212 – what are the implications here? We know that the BBB closes in progressive phases, so what does reduced TJs imply in choroid? A route for immune cell exit/entry during the progressive phase? Is there evidence for this?
224 – the word “primary” here is an assumption. Perhaps say “the structural role … “
240 – the recent Kipnis papers should be mentioned here and paragraph expanded
339 – closed BBB
402 – comma use
422 – is there evidence that switching can happen in B-cells? Reference.
440 – could you elaborate on Nishihara, H., et al. – what are the proposed mechanisms that might cause differences in passage for epithelial VS endothelial?
442 – these are rarely there but complement deposition is widespread. Could you comment on this paradox?
548 - nor was there any
562 - ref Kipnis papers
576 - ref https://www.nature.com/articles/s41593-022-01063-z
582 - s
610 - S1P
655 - s
666 - add refs
711 - binds
Reviewer 4 Report
The review manuscript by Rodriguez-Mogeda et al. with a title “Breaching Brain Barriers: B cell migration in multiple sclerosis” provides a comprehensive overview of the current literature on how the B cells can cross the brain barriers in multiple sclerosis. While most MS related literature focuses on blood-brain barrier and T cell infiltration into the brain, less literature is available on other brain barriers and B cell infiltration routes and their role in the MS pathology. This manuscript therefore closes a knowledge gap by summarizing the relevant literature and describing a “big picture” of this topic. The text is accompanied by clear and understandable figures that aim to visualize the information described in the text. The very interesting part of the review discusses the latest findings on different brain barriers such as blood-brain-, blood-meningeal- and blood cerebrospinal fluid-barriers as possible routes of B cells and other immune cells infiltration into the brain and potential therapeutic targets for multiple sclerosis. In addition, the authors provide a comprehensive overview of the molecular mechanism underlying B cells infiltration into the brain, the barriers and cell types involved, and describe recent clinical strategies to combat MS that target B cells and inhibit their entry into the brain and spinal cord.
It is very helpful to the reader that the authors include a brief summary after each paragraph, pointing out unknown aspects of the B cell migration process and the avenue for future research.
The manuscript is well-written and interesting.
Author Response
We want to thank the reviewer for his/her positive appraisal of our manuscript.